# Trend Feature Consistency Guided Deep Learning Method for Minor Fault Diagnosis

**DOI:** 10.3390/e25020242

**Published:** 2023-01-28

**Authors:** Pengpeng Jia, Chaoge Wang, Funa Zhou, Xiong Hu

**Affiliations:** School of Logistic Engineering, Shanghai Maritime University, Shanghai 201306, China

**Keywords:** deep learning, minor fault, trend feature consistency, fault orientation consistency, small sample size

## Abstract

Deep learning can be applied in the field of fault diagnosis without an accurate mechanism model. However, the accurate diagnosis of minor faults using deep learning is limited by the training sample size. In the case that only a small number of noise-polluted samples is available, it is crucial to design a new learning mechanism for the training of deep neural networks to make it more powerful in feature representation. The new learning mechanism for deep neural networks model is accomplished by designing a new loss function such that accurate feature representation driven by consistency of trend features and accurate fault classification driven by consistency of fault direction both can be secured. In such a way, a more robust and more reliable fault diagnosis model using deep neural networks can be established to effectively discriminate those faults with equal or similar membership values of fault classifiers, which is unavailable for traditional methods. Validation for gearbox fault diagnosis shows that 100 training samples polluted with strong noise are adequate for the proposed method to successfully train deep neural networks to achieve satisfactory fault diagnosis accuracy, while more than 1500 training samples are required for traditional methods to achieve comparative fault diagnosis accuracy.

## 1. Introduction

As a key component of the motor drive system for electric vehicles, a healthy state of the gearbox is critical to the safe operation of autonomous ships, since they are prone to suffer from faults due to heavy loads or mechanical deterioration [1,2,3]. Therefore, fault diagnosis for critical components of the motor drive system is vitally important [4]. However, early minor faults are a challenging diagnostic task, because the fault features are weak and easily submerged by strong noise interference, which makes them difficult to extract and identify [5].

In general, the existing minor fault diagnosis methods can be classified into three categories: physical model-based methods, expert knowledge-based methods, and data-driven methods [6,7]. However, it is difficult to establish accurate physical models for complex systems, which limits the application of physical model-based methods in the engineering field [8]. On the other hand, the expert knowledge-based methods require unique expertise in specific areas, which have limited their generalization [9]. Data-driven methods have received wide attention from engineers since only monitoring data of the operation status are required [10]. As a data-driven approach, deep learning has a good effect on the fault feature extraction of monitoring data. Compared with shallow learning methods, deep learning has the ability to approximate complex functions by means of layer-by-layer feature extraction [11]. According to the difference in network structure, deep learning methods can be classified into four classes: deep belief network (DBN), convolutional neural network (CNN), recurrent neural network (RNN), and stacked auto-encoder (SAE) [12,13,14,15]. Since a 1D vibration signal can be easily collected, the deep neural network (DNN) constructed by SAE is preferred in gearbox fault diagnosis based on deep learning. However, the features of minor faults are generally very weak, and they can be easily buried in strong environmental noise and high-order harmonic components.

The comprehensive extraction capability of minor fault features is affected by strong noise or the limited feature extraction ability of traditional DNNs. The existing DNN-based minor fault diagnosis methods can be classified into three classes: DNN-based minor fault diagnosis method using preprocessing for denoising, DNN-based fault diagnosis method using post-processing for fusion, and fault diagnosis methods using DNN with more powerful feature extraction.

On the aspect of the preprocessing of DNNs, Chen et al. [16] performed Fast Fourier Transform (FFT) to obtain the frequency spectrum of fault signals, and then the frequency spectrum was fed to SAE, which can effectively diagnose faults with weak symptoms in the time domain but significant symptoms in the frequency domain. However, FFT is a linear transform, so it cannot detect minor faults with nonlinear and non-stationary characteristics. Li et al. [17] used Variational Mode Decomposition (VMD) as a preprocessing tool to decompose the original vibration signal with noise into different components. Then, the decomposed signal is fed to SAE, such that it can perform well in the fault diagnosis of a planetary gearbox with insignificant wear faults or others. However, the decomposition level of the VMD algorithm cannot be adaptively selected. Tang et al. [18] used complete ensemble empirical mode decomposition with adaptive noise and Fast Fourier Transform as a preprocessing tool to extract time–frequency features; then, the extracted features are fed to SAE to achieve the accurate diagnosis of minor faults polluted by noise. Although signal preprocessing can alleviate the incapability of diagnosing minor faults submerged by strong noise, the minor fault features may be mis-removed by simply preprocessing the monitoring data. Thus, it is faced with the side effects of DNN-based fault diagnosis using preprocessing.

The problems mentioned above can be avoided by utilizing the post-processing methods of decision-level fusion. On the aspect of post-processing, Jin et al. [19] designed a decision fusion mechanism for multiple feature extraction. Multiple optimization criteria are used to train a unique SDAE model, such that different features involved in a unique set of data are extracted from different points of view. Then, different classification results can be obtained by using these features, and a decision mechanism based on multi-response linear regression is developed to fuse these classification results, which effectively improves the accuracy of planetary gearbox minor fault detection. Zhang et al. [20] proposed a decision fusion mechanism to fuse the classification of CNN and SDAE. For the purpose of the accurate diagnosis of minor faults, D-S evidence theory is used to incorporate a CNN’s capability to process 2D images and SDAE’s capability to process 1D sequences. Although decision-level fusion can effectively solve the problems faced by the preprocessing methods, information loss is serious since the fusion is only performed on the final output of SAE, without taking deep fusion into account.

For this goal, we need to design new networks or learning skills to achieve more powerful DNN feature extraction. Stacked denoising auto-encoder (SDAE) is a commonly used learning tool to alleviate noise effects. On the aspect of powerful feature extraction, Lu et al. [21] used SDAE to decrease noise such that gearbox weak fault features buried by noise can be better recognized. Chen et al. [22] developed an improved SDAE model with moving windows to reconstruct pure operating data from data polluted with different levels of noise. Rashidi [23] proposed a correlative SAE and DNN, which realizes output-related fault diagnosis by building two new constructive and demoting loss functions relatively. Zhu et al. [24] designed a stacked pruning sparse denoising auto-encoder to diagnose minor faults. The developed pruning strategy can eliminate the neurons contributed by pruning the output of SDAE. In such a sense, weak features of minor faults can be better extracted since the non-superior units are pruned out. However, this pruning method may eliminate all information, which is not a satisfactory outcome. Zhou et al. [25] proposed a new DNN structure with a sparse gate to reduce the influence of less contributed neurons, thus achieving minor fault diagnosis with data affected by noise. Although SDAE does well in noise reduction, it is still limited by the poor feature representation capacity without using the idea of information fusion.

The question of how to gain a more comprehensive feature representation using a fusion strategy is significant. Shao et al. [26] developed a fusion mechanism to fuse the features extracted by auto-encoders with different activation functions. The extracted features of two auto-encoders are merged before feeding into the classifier to accurately recognize the minor fault. Kong et al. [27] built several SAEs with different activation functions for feature extraction from different aspects in the first step. They then designed a fusion mechanism with a feature pool to merge the different features, which can be used for minor fault diagnosis. Shi et al. [28] developed a fusion mechanism to fuse the output of a DNN, using the raw data, FFT data, and WPT data as the input of the DNN, respectively. The purpose of fusion is to achieve more comprehensive features from the time domain, frequency domain, and time–frequency domain to recognize minor faults. Zhou et al. [29] developed a deep multi-scale feature fusion mechanism to fuse the features extracted on adjacent layers of a unique SAE. It performs well in minor faults since it can combine features on adjacent layers to compensate for the information missing during the process of layer-by-layer feature extraction. Although Refs. [26,27,28,29] used the same network structure to extract different features by using different fusion mechanisms, a single mode of data is used to fuse features extracted from different aspects, which will inevitably result in feature redundancy. Designing a new learning algorithm to fuse data from different modes is significant in comprehensive feature extraction. Subsequently, Zhou et al. [30] proposed an alternative fusion mechanism to fuse features extracted by SAE to the 1D vibration data and CNN to the 2D image data, respectively. Ravikumar et al. [31] built a multi-scale deep residual learning and stacked LSTM model to achieve gearbox fault diagnosis by fusing features extracted using multiple CNN models and feeding them into a stacked LSTM model. Since this method fully uses heterogeneous data, it can achieve more accurate diagnosis for minor faults.

However, the methods mentioned above cannot effectively extract trend features involved in 1D time series. The fusion of LSTM to trend features and a CNN to static features can be used to solve these problems. Chen et al. [32] proposed an embedded LSTM-CNN auto-encoder to extract trend features that contain both local features and degradation trend information from vibration data. The fused features can be fed into a fault classifier such that minor faults can be well diagnosed. Zhou et al. [33] proposed a fusion network with a new training mechanism to diagnose gearbox minor faults, which is extracted by LSTM to the 1D time series and a CNN to the 2D image data. Although trend features can be extracted by LSTM, it is difficult to train with a large number of network parameters and may suffer from gradient disappearance when dealing with long time series.

This paper focuses on solving the problem encountered by traditional DNNs to diagnose minor faults using a small number of training samples polluted with strong noise. A new training mechanism is established by designing a new loss function where trend feature consistency and fault orientation consistency are both taken into account, such that similar minor faults can be well discriminated. The contributions of this paper are as follows:

1. This paper establishes a new training mechanism for a DNN-based fault diagnosis model to make it more powerful for trend feature representation and the discrimination of similar faults that have equal or similar values of membership.

2. A new loss function for layer-by-by pre-training is designed by taking the consistency of auto-correlation and cross-correlation into account to characterize the trend features involved in the fault signal, such that the extracted features can be more accurate. The loss function considering the consistency of fault orientation is also designed for the backpropagation adjustment of the model to ensure its capability to discriminate similar faults.

3. In the engineering field, when there is only a small number of fault samples polluted by strong noise available, the proposed method is of much significance, since it can provide a more advanced learning mechanism for DNNs to accurately extract the potential weak features of minor faults. Thus, it provides an effective means to secure the accuracy of minor fault diagnosis.

The structure of this paper is as follows. In Section 2, the relevant basic theories are briefly introduced. Section 3 details the specific improvement measures of the proposed method. Section 4 validates the effectiveness and superiority of the proposed method on gearbox and bearing datasets and compared with other mainstream diagnostic techniques. Finally, the conclusions are summarized in Section 5.

## 2. Related Theories

### 2.1. Deep Neural Network Based on Stacked Auto-Encoder

A DNN [34] is built by stacking multiple AEs, and the structure is as shown in Figure 1. An auto-encoder mainly reconstructs the input of the network through the encoding and decoding network. The training process of a DNN is divided into two parts; the first is bottom-up layer-by-layer unsupervised pre-training to obtain a valid feature representation of the original input training data, and this is followed by the top-down supervised global fine-tuning of the training network parameters [35,36].

The corresponding formulas of a forward propagation auto-encoder are as follows [37]:(1)$$\hat{x}=\sigma_{d}(W^{\left(2\right)}\cdot \sigma_{e}(W^{\left(1\right)}x+b^{\left(1\right)}))+b^{\left(2\right)}$$
where $x=\{x_{1},x_{2},\ldots ,x_{m}\}$ denotes the training data; for each sample $x_{i}={\left[x_{i1},x_{i2},\ldots ,x_{iN}\right]}^{T}$, the encoder calculates the input data x to obtain the encoding, and the decoder obtains the reconstruction result ${\hat{x}}_{i}$ of the input data by performing decoding on encoding data. $\hat{x}=\{{\hat{x}}_{1},{\hat{x}}_{2},\ldots ,{\hat{x}}_{m}\}$ is the reconstruction result of the AE, for each reconstruction result ${\hat{x}}_{i}={\left[{\hat{x}}_{i1},{\hat{x}}_{i2},\ldots ,{\hat{x}}_{iN}\right]}^{T}$, and the traditional loss function of feature learning based on amplitude is as follows:(2)$$loss_{MSE}=\frac{1}{m}\sum_{i=1}^{m}\frac{1}{2}{\left\|x_{i}-{\hat{x}}_{i}\right\|}^{2}$$
where m is the number of samples. For a traditional auto-encoder, the reverse derivation of the network model parameters through the chain rule is as shown in Equations (3)–(6).
(3)$$\frac{\partial loss_{MSE}}{\partial W^{\left(2\right)}}=\sum_{i=1}^{m}(x_{i}-{\hat{x}}_{i})\cdot \sigma^{\prime }(Z_{d})\cdot \sigma_{e}(W^{\left(1\right)}x+b^{\left(1\right)})$$
(4)$$W^{\left(2\right)}=W^{\left(2\right)}-\frac{\partial loss_{MSE}}{\partial W^{\left(2\right)}}$$
(5)$$\frac{\partial loss_{MSE}}{\partial b^{\left(2\right)}}=\sum_{i=1}^{m}\left(x_{i}-{\hat{x}}_{i}\right)\cdot \sigma^{\prime }(Z_{d})$$
(6)$$b^{\left(2\right)}=b^{\left(2\right)}-\frac{\partial loss_{MSE}}{\partial b^{\left(2\right)}}$$

### 2.2. Cross-Correlation Coefficient

In signal processing, the cross-correlation coefficient is a measure that describes the degree of correlation between signals and thus enables the identification, detection, and extraction of signals [38], which is defined as shown in Equation (7).
(7)$$\rho_{xy}=\frac{\sum_{i=1}^{n}x_{i}y_{i}}{\sqrt{\sum_{i=1}^{n}x_{i}^{2}\sum_{i=1}^{n}y_{i}^{2}}}$$
where $x_{i}$ and $y_{i}$ are discrete signals, whose length is m. From the Cauchy–Schwarz inequality, $\left|\rho_{xy}\right|\leq 1$, only if $x_{i}=y_{i}$, $\rho_{xy}=1$, and the correlation between $x_{i}$ and $y_{i}$ is the maximum. When $x_{i}$ and $y_{i}$ are correlated, the $\left|\rho_{xy}\right|$ ranges from 0 to 1. On the contrary, when $x_{i}$ is completely independent of $y_{i}$, $\rho_{xy}=0$. The cross-correlation of samples is defined as shown in Equation (8).
(8)$$\rho_{xy}=\frac{\sum_{j=1}^{m}\left[\sum_{i=1}^{n}x_{ij}\cdot y_{ij}\right]}{\sqrt{\sum_{j=1}^{m}\sum_{i=1}^{n}x_{ij}^{2}\sum_{j=1}^{m}\sum_{i=1}^{n}y_{ij}^{2}}}$$
n is the dimension of the sample, and m is the number of samples.

### 2.3. Auto-Correlation Coefficient

The auto-correlation coefficient of a random signal reflects the degree of correlation of the signal itself at different times [39]. The auto-correlation coefficient is defined as shown in Equation (9).
(9)$$\gamma_{h}=\frac{\mathrm{cov}(x_{i},x_{i-h})}{\sqrt{D(x_{i})D(x_{i-h})}},h=1,2,\cdots$$
where h is the order of the auto-correlation coefficient and $D\left(x_{i}\right)$ is the variance of x. The set of $\left\{\gamma_{h}\right\}$ auto-correlation coefficients is called the auto-correlation function. In probability and statistical parameter estimation, for the overall sample $\left\{x_{t}\right\}$, the variance and covariance all contain unknown parameters. Therefore, it is necessary to rely on the sample auto-correlation coefficient. For a given set of samples $x=\left\{x_{1},x_{2},\ldots ,x_{m}\right\}$, the sample auto-correlation coefficient with interval h is defined as shown in Equation (10).
(10)$$\gamma_{h}=\sum_{i=1}^{n}\frac{\sum_{j=h+1}^{m}\left(x_{ji}-\bar{x_{i}}\right)(x_{(j-h)i}-\bar{x_{i}})}{\sum_{j=1}^{m}{\left(x_{ji}-\bar{x_{i}}\right)}^{2}},0\leq h\leq m-1$$
where $\bar{x}$ is the mean of x. n is the dimension of the sample, and m is the number of samples.

### 2.4. Cosine Distance

In geometry, the angle cosine is used to measure the difference between the directions of two vectors. In machine learning, this concept is used to measure differences between sample vectors. Compared with distance measurement, the cosine distance pays more attention to the difference in direction between two vectors, rather than the difference in distance or length [40]. The cosine distance in m-dimensional space is
(11)$$\cos({\vec{y}}_{pre},{\vec{y}}_{true})=\frac{\sum_{i=1}^{C}{\vec{y}}_{pre_{i}}{\vec{y}}_{real_{i}}}{\sqrt{\sum_{i=1}^{C}{\vec{y}}_{pre_{i}}^{2}}\sqrt{\sum_{i=1}^{C}{\vec{y}}_{real_{i}}^{2}}}$$
where ${\vec{y}}_{pre}$ is the prediction label, and ${\vec{y}}_{real}$ is the real label. This cosine value can be used to represent the similarity of the two vectors. The smaller the included angle, the closer to 0 degrees, the closer the cosine value is to 1, the more consistent their directions are, and the more similar they are. When the directions of the two vectors are completely opposite, the cosine of the included angle takes the minimum value of −1. When the cosine value is 0, the two vectors are orthogonal and the included angle is 90°. Therefore, it can be seen that the cosine similarity is not related to the magnitude of the vector, but only to the direction of the vector, where the vectors ${\vec{y}}_{pre}$ and ${\vec{y}}_{train}$ are one-dimensional vectors. Therefore, the cosine distance in the one-dimensional vector form is generalized to the two-dimensional matrix form, as follows:(12)$$\cos(y_{pre},y_{true})=\sum_{j=1}^{n}\frac{\sum_{i=1}^{C}y_{pre_{ij}}\cdot y_{real_{ij}}}{\sqrt{\sum_{i=1}^{C}y^{2}{}_{pre_{ij}}}\sqrt{\sum_{i=1}^{C}y^{2}{}_{real_{ij}}}}$$
where $y_{pre}$ and $y_{real}$ are $R^{C\times n}$, and C is the total number of labels. n is the sample size.

## 3. Minor Fault Diagnosis Method Using DNN Guided by Consistency of Trend Features and Consistency of Fault Orientation

### 3.1. Incapability of DNN to Extract Separable Features of Similar Faults

The basic idea of the traditional AE learning process is to ensure that the features obtained by the coding network can reconstruct the input data of the encoder. Its optimization criterion is to construct the loss function only by the amplitude consistency of the signal, but there may still be trend differences between the input and output, which leads to the loss of the trend features of the faulty signal during the representation. However, the lost trend information is key to minor fault diagnosis. Figure 2 shows that a minor fault is difficult to diagnose because the fault feature extracted by using the current learning mechanism is similar to other features.

In Figure 2, the solid black line represents a normal signal F_0_. The cyan dashed line represents the first fault signal F_1_, which is generated by subtracting a slowly drifting signal from F_0_. The magenta dotted line represents the second fault signal F_2_, which is constructed by subtracting the slowly drifting signal for an odd sample time of F_0_ and adding the slowly drifting signal at an even sample time of F_0_. The dash-dotted red line represents the third fault signal F_3_, which is generated by adding the slowly drifting signal at an odd sample time of F_0_ and subtracting the slowly drifting signal at an even sample time of F_0_. It can be easily seen that the loss defined only by amplitude consistency is equal in $loss_{MSE}(F_{0},F_{1}),loss_{MSE}(F_{0},F_{2})$, and $loss_{MSE}(F_{0},F_{3})$, which means that it is difficult to discriminate the three similar faults mentioned above. On the other hand, Figure 2 shows that the trend features of $F_{0}$, $F_{2}$, and $F_{3}$ are completely different, which means that they are three different types of faults. Thus, the question of how to design a new training mechanism to extract potential trend features is crucial to minor fault diagnosis.

If the features of the three faults mentioned above are not well represented, it will lead to fault classification errors due to the proximity of the first and second memberships in the fault classification. Figure 3 shows that it is possible to misclassify similar faults if the method is incapable of obtaining an adequate difference between the first and the second membership. In Figure 3, the red dots indicate the first membership and the green dots indicate the second membership.

### 3.2. Minor Fault Diagnosis Using DNN Guided by Consistency of Trend Features and Consistency of Fault Orientation

#### 3.2.1. Trend Feature-Guided Training of a Unique AE

This section focuses on developing a new training mechanism by designing a new loss function to extract more accurate features by the AE. Rather than defining the loss function related to amplitude consistency, we choose to design a new loss function taking both amplitude consistency and trend feature consistency into account, such that additional trend features that are helpful for minor fault diagnosis can be well extracted. As shown in Figure 4, the auto-correlation for the input signal of the AE and the cross-correlation between the input and output of the AE (CASAE) are used to characterize the trend feature.

The main idea is to minimize the difference between the input and output of the AE by using the loss function designed in Equation (13).
(13)$$\begin{array}{cc} loss(W_{en},b_{en},W_{de},b_{de}) & ={loss}_{MSE}+{loss}_{c\text{\_}c}+loss_{a\text{\_}c}\\ & =\frac{1}{2m}\sum_{i=1}^{m}{\left\|x_{i}-{\hat{x}}_{i}\right\|}^{2}+\frac{1}{m}{\sum_{i=1}^{m}\left\|1-\rho (x_{i},{\hat{x}}_{i})\right\|}^{2}+\frac{1}{2lages}\sum_{h=1}^{lags}{\left\|\gamma_{h}(x(t),x(t-h))-\gamma_{h}(\hat{x}(t),\hat{x}(t-h))\right\|}^{2} \end{array}$$
where $W_{en},b_{en}$ are the weight and bias matrix of the encoder, respectively. $W_{de},b_{de}$ are the weight and bias matrix of the decoder, respectively. $x_{i}$ is the input data of the AE, and ${\hat{x}}_{i}$ is the reconstructed result of the AE. The cross-correlation coefficient is used to capture the trend feature reconstruction error index $\rho \left(x,\hat{x}\right)=\frac{cov\left(x,\hat{x}\right)}{s\left(x\right)s\left(\hat{x}\right)}$, where $cov\left(x,\hat{x}\right)$ is the covariance matrix of *x* and $\hat{x}$, and $s\left(x\right)$ is the standard deviation of x. Another reconstruction error index $\gamma_{h}\left(x\left(t\right),x\left(t-h\right)\right)=\frac{\left(x_{i}-\overset{¯}{x}\right)\left(x_{i-h}-\overset{¯}{x}\right)}{s^{2}\left(x\right)}$, $s^{2}\left(x\right)$ is the variance of x, and h is the order of the auto-correlation coefficient, which is also used to extract the trend feature.

$loss_{c\text{\_}c}$ aims to extract the trend feature of the correlation between the input and output of the AE, which can be defined as in Equation (14).
(14)$$\begin{array}{cc} loss_{c\text{\_}c} & =\frac{1}{m}{\sum_{i=1}^{m}\left\|1-\rho (x_{i},{\hat{x}}_{i})\right\|}^{2}\\ & =\frac{1}{m}{\sum_{i=1}^{m}\left\|1-\frac{\frac{1}{m}\sum_{i=1}^{m}\left(x_{i}-\frac{1}{m}\sum_{i=1}^{m}x_{i})({\hat{x}}_{i}-\frac{1}{m}\sum_{i=1}^{m}{\hat{x}}_{i}\right)}{\sqrt{\frac{1}{m}\sum_{i=1}^{m}{\left(x_{i}-\frac{1}{m}\sum_{i=1}^{m}x_{i}\right)}^{2}}\sqrt{\frac{1}{m}\sum_{i=1}^{m}{\left({\hat{x}}_{i}-\frac{1}{m}\sum_{i=1}^{m}{\hat{x}}_{i}\right)}^{2}}}\right\|}^{2} \end{array}$$

From Equation (14), the trend consistency for the input and output of data is characterized by cross-correlation. The optimization criterion for minimizing the reconstruction error in the feature process can be achieved when the trend features for the input and output of the AE are consistent.

$loss_{a\text{\_}c}$ aims to extract the trend distribution features for the input and output of AE data and can be defined as in Equation (15).
(15)$$\begin{array}{cc} loss_{a\text{\_}c} & =\frac{1}{2lages}\sum_{h=1}^{lags}{\left\|\gamma_{h}(x(t),x(t-h))-\gamma_{h}(\hat{x}(t),\hat{x}(t-h))\right\|}^{2}\\ & =\frac{1}{2lages}{\sum_{j=1}^{N}\left\|\sum_{h=1}^{lags}\left(\frac{\sum_{i=h+1}^{m}\left(x_{ij}-{\bar{x}}_{ij})(x_{(i-h)j}-{\bar{x}}_{ij}\right)}{\sum_{i=1}^{m}{\left(x_{ij}-{\bar{x}}_{ij}\right)}^{2}}-\frac{\sum_{i=h+1}^{m}\left({\hat{x}}_{ij}-{\bar{\hat{x}}}_{ij})({\hat{x}}_{(i-h)j}-{\bar{\hat{x}}}_{ij}\right)}{\sum_{i=1}^{m}{\left({\hat{x}}_{ij}-{\bar{\hat{x}}}_{ij}\right)}^{2}}\right)\right\|}^{2} \end{array}$$

From Equation (15), if the input and output data of the AE are identical, then their cross-correlation and auto-correlation are equal as well. In the case that the amplitude difference is consistent, $loss_{a\text{\_}c}$ in Equation (13) can well capture the difference in trend features for the input and output of the AE.

The detailed backpropagation algorithm corresponding to the new loss function defined in Equations (16)–(19) is as follows:(16)$$W_{de}=W_{de}-lr\times (\frac{\partial loss_{MSE}}{\partial W_{de}}+\frac{\partial loss_{c\text{\_}c}}{\partial W_{de}}+\frac{\partial loss_{a\text{\_}c}}{\partial W_{de}})$$
(17)$$b_{de}=b_{de}-lr\times (\frac{\partial loss_{MSE}}{\partial b_{de}}+\frac{\partial loss_{c\text{\_}c}}{\partial b_{de}}+\frac{\partial loss_{a\text{\_}c}}{\partial b_{de}})$$
(18)$$W_{en}=W_{en}-lr\times (\frac{\partial loss_{MSE}}{\partial W_{en}}+\frac{\partial loss_{c\text{\_}c}}{\partial W_{en}}+\frac{\partial loss_{a\text{\_}c}}{\partial W_{en}})$$
(19)$$b_{en}=b_{en}-lr\times (\frac{\partial loss_{MSE}}{\partial b_{en}}+\frac{\partial loss_{c\text{\_}c}}{\partial b_{en}}+\frac{\partial loss_{a\text{\_}c}}{\partial b_{en}})$$
where lr is the learning rate; $\frac{\partial loss_{MSE}}{\partial *}$, $\frac{\partial loss_{c\text{\_}c}}{\partial *}$, and $\frac{\partial loss_{a\text{\_}c}}{\partial *}$ are, respectively, the loss functions constructed as mentioned above. * corresponds to each parameter of the AE. The gradient for $loss_{MSE}$ is calculated as follows:(20)$$\frac{\partial loss_{MSE}}{\partial W_{de}}=\frac{1}{m}\sum_{i=1}^{m}(x_{i}-{\hat{x}}_{i})\cdot {\sigma^{\prime }}_{de}(z_{de})\cdot \sigma_{en}(z_{en})$$
(21)$$\frac{\partial loss_{MSE}}{\partial b_{de}}=\frac{1}{m}\sum_{i=1}^{m}\left(x_{i}-{\hat{x}}_{i}\right)\cdot {\sigma^{\prime }}_{de}(z_{de})$$
(22)$$\frac{\partial loss_{MSE}}{\partial W_{en}}=\frac{1}{m}\sum_{i=1}^{m}(x_{i}-{\hat{x}}_{i})\cdot {\sigma^{\prime }}_{de}(z_{de})\cdot W_{de}\cdot {\sigma^{\prime }}_{en}(z_{en})\cdot x$$
(23)$$\frac{\partial loss_{MSE}}{\partial W_{en}}=\frac{1}{m}\sum_{i=1}^{m}(x_{i}-{\hat{x}}_{i})\cdot {\sigma^{\prime }}_{de}(z_{de})\cdot W_{de}\cdot {\sigma^{\prime }}_{en}(z_{en})$$

The gradient for $loss_{c\text{\_}c}$ is calculated as follows:(24)$$\frac{\partial loss_{c\text{\_}c}}{\partial W_{de}}=\frac{1}{m}\sum_{i=1}^{m}\left(1-\frac{\frac{1}{m}\sum_{i=1}^{m}\left(x_{i}-\bar{x}\right)({\hat{x}}_{i}-\frac{1}{m}\sum_{i=1}^{m}{\hat{x}}_{i})}{\sqrt{\frac{1}{m^{2}}\sum_{i=1}^{m}{\left(x_{i}-\bar{x}\right)}^{2}\sum_{i=1}^{m}{\left({\hat{x}}_{i}-\frac{1}{m}\sum_{i=1}^{m}{\hat{x}}_{i}\right)}^{2}}}\right){\left(-\frac{\frac{1}{m}\sum_{i=1}^{m}\left(x_{i}-\bar{x}\right)({\hat{x}}_{i}-\frac{1}{m}\sum_{i=1}^{m}{\hat{x}}_{i})}{\sqrt{\frac{1}{m^{2}}\sum_{i=1}^{m}{\left(x_{i}-\bar{x}\right)}^{2}\sum_{i=1}^{m}{\left({\hat{x}}_{i}-\frac{1}{m}\sum_{i=1}^{m}{\hat{x}}_{i}\right)}^{2}}}\right)}^{\prime }{\sigma^{\prime }}_{de}(z_{de})\sigma_{en}(z_{en})$$
(25)$$\frac{\partial loss_{c\text{\_}c}}{\partial b_{de}}=\frac{1}{m}\sum_{i=1}^{m}\left(1-\frac{\frac{1}{m}\sum_{i=1}^{m}\left(x_{i}-\bar{x}\right)({\hat{x}}_{i}-\frac{1}{m}\sum_{i=1}^{m}{\hat{x}}_{i})}{\sqrt{\frac{1}{m^{2}}\sum_{i=1}^{m}{\left(x_{i}-\bar{x}\right)}^{2}\sum_{i=1}^{m}{\left({\hat{x}}_{i}-\frac{1}{m}\sum_{i=1}^{m}{\hat{x}}_{i}\right)}^{2}}}\right)(-\frac{\frac{1}{m}\sum_{i=1}^{m}\left(x_{i}-\bar{x}\right)({\hat{x}}_{i}-\frac{1}{m}\sum_{i=1}^{m}{\hat{x}}_{i})}{\sqrt{\frac{1}{m^{2}}\sum_{i=1}^{m}{\left(x_{i}-\bar{x}\right)}^{2}\sum_{i=1}^{m}{\left({\hat{x}}_{i}-\frac{1}{m}\sum_{i=1}^{m}{\hat{x}}_{i}\right)}^{2}}}{)}^{\prime }{\sigma^{\prime }}_{de}(Z_{de})$$
(26)$$\frac{\partial loss_{c\text{\_}c}}{\partial W_{en}}=\frac{1}{m}\sum_{i=1}^{m}\left(1-\frac{\frac{1}{m}\sum_{i=1}^{m}\left(x_{i}-\bar{x}\right)\cdot ({\hat{x}}_{i}-\frac{1}{m}\sum_{i=1}^{m}{\hat{x}}_{i})}{\sqrt{\frac{1}{m^{2}}\sum_{i=1}^{m}{\left(x_{i}-\bar{x}\right)}^{2}\sum_{i=1}^{m}{\left({\hat{x}}_{i}-\frac{1}{m}\sum_{i=1}^{m}{\hat{x}}_{i}\right)}^{2}}}\right)\cdot (-\frac{\frac{1}{m}\sum_{i=1}^{m}\left(x_{i}-\bar{x}\right)\cdot ({\hat{x}}_{i}-\frac{1}{m}\sum_{i=1}^{m}{\hat{x}}_{i})}{\sqrt{\frac{1}{m^{2}}\sum_{i=1}^{m}{\left(x_{i}-\bar{x}\right)}^{2}\sum_{i=1}^{m}{\left({\hat{x}}_{i}-\frac{1}{m}\sum_{i=1}^{m}{\hat{x}}_{i}\right)}^{2}}}{)}^{\prime }\cdot {\sigma^{\prime }}_{de}(z_{de})\cdot W_{de}\cdot {\sigma^{\prime }}_{en}(z_{en})\cdot x$$
(27)$$\frac{\partial loss_{c\text{\_}c}}{\partial b_{en}}=\frac{1}{m}\sum_{i=1}^{m}\left(1-\frac{\frac{1}{m}\sum_{i=1}^{m}\left(x_{i}-\bar{x}\right)({\hat{x}}_{i}-\frac{1}{m}\sum_{i=1}^{m}{\hat{x}}_{i})}{\sqrt{\frac{1}{m^{2}}\sum_{i=1}^{m}{\left(x_{i}-\bar{x}\right)}^{2}\sum_{i=1}^{m}{\left({\hat{x}}_{i}-\frac{1}{m}\sum_{i=1}^{m}{\hat{x}}_{i}\right)}^{2}}}\right)(-\frac{\frac{1}{m}\sum_{i=1}^{m}\left(x_{i}-\bar{x}\right)({\hat{x}}_{i}-\frac{1}{m}\sum_{i=1}^{m}{\hat{x}}_{i})}{\sqrt{\frac{1}{m^{2}}\sum_{i=1}^{m}{\left(x_{i}-\bar{x}\right)}^{2}\sum_{i=1}^{m}{\left({\hat{x}}_{i}-\frac{1}{m}\sum_{i=1}^{m}{\hat{x}}_{i}\right)}^{2}}}{)}^{\prime }{\sigma^{\prime }}_{de}(z_{de})W_{de}{\sigma^{\prime }}_{en}(z_{en})$$

The gradient for $loss_{a\text{\_}c}$ is calculated as follows:(28)$$\frac{\partial loss_{a\text{\_}c}}{\partial W_{de}}=\frac{1}{lags}\sum_{j=1}^{N}\left[\sum_{h=1}^{lags}\left(\frac{\sum_{i=h+1}^{m}\left(x_{ij}-{\bar{x}}_{ij})(x_{(i-h)j}-{\bar{x}}_{ij}\right)}{\sum_{i=1}^{m}{\left(x_{ij}-{\bar{x}}_{ij}\right)}^{2}}-\frac{\sum_{i=h+1}^{m}\left({\hat{x}}_{ij}-{\bar{\hat{x}}}_{ij})({\hat{x}}_{(i-h)j}-{\bar{\hat{x}}}_{ij}\right)}{\sum_{i=1}^{m}{\left({\hat{x}}_{ij}-{\bar{\hat{x}}}_{ij}\right)}^{2}}\right)\right](-\frac{\sum_{i=h+1}^{m}\left({\hat{x}}_{ij}-{\bar{\hat{x}}}_{ij})({\hat{x}}_{(i-h)j}-{\bar{\hat{x}}}_{ij}\right)}{\sum_{i=1}^{m}{\left({\hat{x}}_{ij}-{\bar{\hat{x}}}_{ij}\right)}^{2}}{)}^{\prime }{\sigma^{\prime }}_{de}(z_{de})\sigma_{en}(z_{en})$$
(29)$$\frac{\partial loss_{a\text{\_}c}}{\partial b_{de}}=\frac{1}{lags}\sum_{j=1}^{N}\left[\sum_{h=1}^{lags}\left(\frac{\sum_{i=h+1}^{m}\left(x_{ij}-{\bar{x}}_{ij})(x_{(i-h)j}-{\bar{x}}_{ij}\right)}{\sum_{i=1}^{m}{\left(x_{ij}-{\bar{x}}_{ij}\right)}^{2}}-\frac{\sum_{i=h+1}^{m}\left({\hat{x}}_{ij}-{\bar{\hat{x}}}_{ij})({\hat{x}}_{(i-h)j}-{\bar{\hat{x}}}_{ij}\right)}{\sum_{i=1}^{m}{\left({\hat{x}}_{ij}-{\bar{\hat{x}}}_{ij}\right)}^{2}}\right)\right](-\frac{\sum_{i=h+1}^{m}\left({\hat{x}}_{ij}-{\bar{\hat{x}}}_{ij})({\hat{x}}_{(i-h)j}-{\bar{\hat{x}}}_{ij}\right)}{\sum_{i=1}^{m}{\left({\hat{x}}_{ij}-{\bar{\hat{x}}}_{ij}\right)}^{2}}{)}^{\prime }{\sigma^{\prime }}_{de}(Z_{de})$$
(30)$$\frac{\partial loss_{a\text{\_}c}}{\partial W_{en}}=\frac{1}{lags}\sum_{j=1}^{N}\left[\sum_{h=1}^{lags}\left(\frac{\sum_{i=h+1}^{m}\left(x_{ij}-{\bar{x}}_{ij})(x_{(i-h)j}-{\bar{x}}_{ij}\right)}{\sum_{i=1}^{m}{\left(x_{ij}-{\bar{x}}_{ij}\right)}^{2}}-\frac{\sum_{i=h+1}^{m}\left({\hat{x}}_{ij}-{\bar{\hat{x}}}_{ij})({\hat{x}}_{(i-h)j}-{\bar{\hat{x}}}_{ij}\right)}{\sum_{i=1}^{m}{\left({\hat{x}}_{ij}-{\bar{\hat{x}}}_{ij}\right)}^{2}}\right)\right](-\frac{\sum_{i=h+1}^{m}\left({\hat{x}}_{ij}-{\bar{\hat{x}}}_{ij})({\hat{x}}_{(i-h)j}-{\bar{\hat{x}}}_{ij}\right)}{\sum_{i=1}^{m}{\left({\hat{x}}_{ij}-{\bar{\hat{x}}}_{ij}\right)}^{2}}{)}^{\prime }{\sigma^{\prime }}_{de}(z_{de})W_{de}{\sigma^{\prime }}_{en}(z_{en})x$$
(31)$$\frac{\partial loss_{a\text{\_}c}}{\partial b_{en}}=\frac{1}{lags}\sum_{j=1}^{N}\left[\sum_{h=1}^{lags}\left(\frac{\sum_{i=h+1}^{m}\left(x_{ij}-{\bar{x}}_{ij})(x_{(i-h)j}-{\bar{x}}_{ij}\right)}{\sum_{i=1}^{m}{\left(x_{ij}-{\bar{x}}_{ij}\right)}^{2}}-\frac{\sum_{i=h+1}^{m}\left({\hat{x}}_{ij}-{\bar{\hat{x}}}_{ij})({\hat{x}}_{(i-h)j}-{\bar{\hat{x}}}_{ij}\right)}{\sum_{i=1}^{m}{\left({\hat{x}}_{ij}-{\bar{\hat{x}}}_{ij}\right)}^{2}}\right)\right](-\frac{\sum_{i=h+1}^{m}\left({\hat{x}}_{ij}-{\bar{\hat{x}}}_{ij})({\hat{x}}_{(i-h)j}-{\bar{\hat{x}}}_{ij}\right)}{\sum_{i=1}^{m}{\left({\hat{x}}_{ij}-{\bar{\hat{x}}}_{ij}\right)}^{2}}{)}^{\prime }{\sigma^{\prime }}_{de}(z_{de})W_{de}{\sigma^{\prime }}_{en}(z_{en})$$
where ${\sigma^{\prime }}_{de}(z_{de})=1-{\left[\sigma_{de}(z_{de})\right]}^{2}$, ${\sigma^{\prime }}_{en}(z_{en})=1-{\left[\sigma_{en}(z_{en})\right]}^{2}$ denote the tanh activation function. The total gradient of weights and bias can be calculated via Equations (32)–(35):(32)$$\frac{\partial loss}{\partial W_{de}}=\frac{\partial loss_{MSE}}{\partial W_{de}}+\frac{\partial loss_{c\text{\_}c}}{\partial W_{de}}+\frac{\partial loss_{a\text{\_}c}}{\partial W_{de}}$$
(33)$$\frac{\partial loss}{\partial b_{de}}=\frac{\partial loss_{MSE}}{\partial b_{de}}+\frac{\partial loss_{c\text{\_}c}}{\partial b_{de}}+\frac{\partial loss_{a\text{\_}c}}{\partial b_{de}}$$
(34)$$\frac{\partial loss}{\partial W_{en}}=\frac{\partial loss_{MSE}}{\partial W_{en}}+\frac{\partial loss_{c\text{\_}c}}{\partial W_{en}}+\frac{\partial loss_{a\text{\_}c}}{\partial W_{en}}$$
(35)$$\frac{\partial loss}{\partial b_{en}}=\frac{\partial loss_{MSE}}{\partial b_{en}}+\frac{\partial loss_{c\text{\_}c}}{\partial b_{en}}+\frac{\partial loss_{a\text{\_}c}}{\partial b_{en}}$$

#### 3.2.2. Orientation Consistency-Guided Training of SAE in Stage of Backpropagation

The training of SAE involves a supervised global fine-tuning process with an additional classifier on the top layer of the DNN. If similar faults, as shown in Figure 3, occur in the system, the traditional training mechanism is unable to extract separable features for them, as the membership of the classifier is guided by amplitude consistency without considering the consistency of fault orientation.

In order to solve the above-mentioned problem, this section designs a fault orientation consistency-guided training mechanism for SAE. Aiming to discriminate between the similar faults shown in Figure 2, amplitude consistency and fault orientation consistency are used to guide the global adjustment of the backpropagation stage. A schematic diagram of the training mechanism guided by fault orientation consistency for the global adjustment of the DNN (G-CASAE) is shown in Figure 5.

This section focuses on designing a new classification mechanism for SAE, in the event that the fault features are similar, by taking both amplitude consistency and fault orientation consistency into account, to address the problem that the first and second membership of predicted labels are similar. The main idea is to minimize the difference between the predicted labels and the real labels by using the loss function designed in Equation (36).
(36)$$\begin{array}{cc} loss_{a\text{\_}o} & =loss_{dis}+loss_{orn}\\ & =\frac{1}{m}\sum_{i=1}^{m}\frac{1}{2}||y_{pre_{i}}-y_{real_{i}}|{|}^{2}+\frac{1}{m}\sum_{i=1}^{m}\frac{1}{2}||1-\frac{y_{pre_{i}}\cdot y_{real_{i}}}{\left|y_{pre_{i}}\right|\left|y_{real_{i}}\right|}|{|}^{2} \end{array}$$
where $y_{pre}$ is the predicted label, and $y_{real}$ is the real label.

$loss_{orn}$ aims to the extract the orientation difference in the predicted labels and the real labels in Equation (37).
(37)$$loss_{orn}=\frac{1}{m}\sum_{i=1}^{m}\frac{1}{2}||1-\frac{y_{pre_{i}}\cdot y_{real_{i}}}{\left|y_{pre_{i}}\right|\left|y_{real_{i}}\right|}|{|}^{2}$$
where $\frac{y_{pre_{i}}\cdot y_{real_{i}}}{\left|y_{pre_{i}}\right|\left|y_{real_{i}}\right|}$ is used to measure the orientation consistency between the predicted label and the real label in the case in which the amplitude consistency is zero.

#### 3.2.3. Trend Feature Consistency-Driven Deep Learning for Minor Fault Diagnosis

When the online data $x_{online}(t)$ are collected, they are fed into the trained deep neural network with global fine-tuning (G-CASAE) for feature extraction. $Feature_{CF}(t)$ indicates the features used for classification, which are extracted from the online data by the trained model, as shown in Equation (38).
(38)$$Feature_{CF}(t)=G_{CF}(Net_{G-CASAE},Tr_{global},x_{online}(t))$$
where $Net_{G-CASAE}$ is the network model structure, and $Tr_{global}$ are the trained model parameters. The feature $Feature_{CF}\left(t\right)$ is fed into the classifier for fault diagnosis, as shown in Equations (39) and (40).
(39)$$\begin{array}{cc} M_{\theta ,G-CASAE}(t) & =\left[\begin{array}{c} p(label=1)|(Feature_{CF}(t);\theta_{classifier})\\ p(label=2)|(Feature_{CF}(t);\theta_{classifier})\\ \vdots \\ p(label=L)|(Feature_{CF}(t);\theta_{classifier}) \end{array}\right]\\ & =\frac{1}{\sum_{l=1}^{L}\theta_{f_{l}}^{T}Feature_{CF}(t)}\left[\begin{array}{l} e^{\theta_{f_{1}}^{T}Feature_{CF}(t)}\\ e^{\theta_{f_{2}}^{T}Feature_{CF}(t)}\\ \;\vdots \\ e^{\theta_{f_{L}}^{T}Feature_{CF}(t)} \end{array}\right] \end{array}$$
(40)$$label[x_{online}(t)]=\underset{m=1,2,\cdots ,L}{\mathrm{argmax}}\left\{M_{\theta_{classifier},G-CASAE}(t)|(x_{online}(t);\theta_{classifier})\right\}$$
where $\theta_{classifier}$ is the trained parameter of the classifier, and $M_{\theta ,G-CASAE}\left(t\right)$ is the probability that the online sample belongs to each category. $label\left[x_{online}\left(t\right)\right]$ is the fault diagnosis result at time t for the online data. The specific steps of the algorithm are described as follows (Algorithm 1).
**Algorithm 1: Gradient Descent Algorithm for Training G-CASAE with Constructive Loss Function****Step** **1:**The data acquisition system collects the multivariable monitoring signals of the key components of the rotating machinery.**Step** **2:**Prepare the SAE for fault diagnosis.**Step** **3:**Set the parameters of SAE, including the number of neurons for each layer, the learning rate, and the maximum generation number or threshold for exit training.**Step** **4:**Initialize all parameters of each layer W,b to be learned by backpropagation.**Step** **5:**Amplitude consistency and trend consistency are used to design the new SAE loss function for layer-by-layer feature extraction to extract the weak fault trend feature, which aims to enhance the feature extraction capability from the monitoring signals.**Step** **6:**The well-learned features are fed into the classification layer, which is classified by the loss function constructed with amplitude distance consistency and orientation consistency.**Step** **7:**Save the network model parameters.**Step** **8:**Extraction and diagnosis of minor fault features using the trained model.**Step** **9:**Output fault diagnosis results.

A flowchart of the feature extraction process of the new learning method is shown in the following Figure 6.

## 4. Experimental Analysis

Rolling bearings and gearboxes are important components in the motor drive system for autonomous ships. In this section, the effectiveness of the proposed model algorithm is verified using a gearbox dataset and rolling bearing dataset.

### 4.1. Fault Diagnosis for Parallel Gearbox

#### 4.1.1. Dataset 1: Gearbox Data

The gearbox data obtained from the platform of QPZZ-II rotary machinery vibration are used as the benchmark data to test the efficiency of the proposed method [41]. The experimental platform is shown in Figure 7. It mainly consists of a speed drive motor, bearing, parallel gearbox, governor, and so on. The structural parameters of the gearbox are as follows: the gear module is 2 mm, the number of large gear teeth is 75, and the number of small gear teeth is 55. During the experiment, the drive motor was run at 880 rpm, with a braking torque output current of 0.2 A. The gears were cut on either one or both sides by means of wire cutting to simulate faults of broken teeth and wear faults. The pitting fault was simulated by the EDM technique. Healthy monitoring data of the gearbox were collected for H: normal, F1: wear, F2: pitting, F3: broken tooth, F4: pitting and wear fault, respectively. The layout of the sensors to collect data is shown in Table 1, and the sampling frequency was 5120 Hz. The raw data of the collected monitoring signals from 9 channels were fed directly into the proposed model to identify the health status of the gearbox.

In Experiment 1, under different fault conditions, five sets of gearbox data corresponding to different fault conditions were collected. Table 2 shows the experimental scenarios for Experiment 1.

To verify the superiority of the proposed method, five other minor fault diagnosis methods were compared, as shown in Table 3.

#### 4.1.2. Results and Analysis of Experiment 1

The gearbox data are not subject to any preprocessing and artificial feature extraction, as we aim to use the original data for fault diagnosis. When the network models are constructed, the influences of the neuron numbers in each layer are considered. Firstly, the number of neurons in the input layer depends on the dimension of the input sample. Secondly, the setting of the neuron number in the hidden layer follows the principle of first upgrading the input data to a high-dimensional space, and then compressing the high-dimensional features in turn. Therefore, the designed model structure and the compared models are as shown in Table 4.

Simultaneously, in order to illustrate the superiority of the proposed G-CASAE algorithm, it is compared with five other methods. The diagnostic accuracy of each neural network model with a training sample size of 100 in Experiment 1 is given in Table 5.

The fault diagnosis result with a training sample size of 100 is shown in Table 5. As can be seen from columns 2 to 4 in row 2, when the training samples are small, deep learning can be less effective than shallow learning for fault diagnosis due to overfitting. However, BP is unable to mine the intrinsic features of the collected data under the influence of strong noise. Therefore, it is still necessary to use the deep learning method for fault diagnosis. Comparing columns 4 and 5 in row 2, it can be seen that the correlation-based LSTM fault diagnosis model has a significant advantage in its signal trend feature extraction capability, and it can effectively alleviate the overfitting problem of SAE and SDAE with small samples as well. Comparing columns 5 and 6 in row 2, MSFSAE can effectively alleviate the problems of poor diagnosis due to small samples and inconspicuous fault features. It is better than the traditional method of using only the last layer of features for fault diagnosis, but there is also a redundancy feature. Therefore, from the contrastive analysis of columns 6 and 7 in row 2, the proposed CASAE method not only considers the consistency of amplitude in the process of feature extraction, but also uses the consistency of trends to extract the fault features of signals. Compared with MSFSAE, the designed method can effectively guide the learning parameters in the unsupervised pre-training process toward the optimal parameters with relevant feature extraction, which eliminates the interference of redundant features. The comparison of columns 7 and 8 in row 2 shows that when the label similarity measure is added to the global fine-tuning loss function of SAE, the proposed method G-CASAE can effectively solve the problem of misclassification caused by the similar membership of predicted labels and improve the accuracy of fault diagnosis.

The comparative analysis of rows 2 to 5 in Table 5 shows that the diagnostic accuracy decreases with the increase in noise, but the proposed method G-CASAE still has good robustness. Comparing columns 6 and 8 of row 5 in Table 5 indicates that the accuracy of the proposed G-CASAE method is 32.5% higher than the feature fusion MSFSAE method when the effect is stronger. In Table 6 to illustrate the traditional method requires large samples to achieve the di-agnostic accuracy of the proposed method at small samples with signal to noise ratio of 20 dB.

Through the analysis with a signal to noise ratio of 20 dB and a sample size of 1500 in Table 6, it can be found that when the sample size is large enough, the five traditional methods can achieve similar diagnostic accuracy to the designed method with small samples. Therefore, the fault diagnosis effect of the proposed algorithm is obviously superior to the traditional fault diagnosis method under small samples and strong noise.

To summarize, the experimental results show that the designed G-CASAE method has better performance and stronger noise robustness than other methods. To further visualize the diagnostic effect, the confusion matrix for each type of fault diagnosis is shown in Figure 8. The darker color indicates that more samples are correctly diagnosed for each type of data in Figure 8.

To further illustrate the superiority and robustness of the proposed method, the confusion matrix of these methods is given in Figure 8. As we can see in Figure 8, the proposed G-CASAE has the highest number of correct diagnostic samples for all fault types. It is clear from the diagnostic results of the seven confusion matrices that there is a mutual misclassification problem in the second and third rows. From Figure 8a–d, the other five compared models have the most misclassifications in pitting–wear composite faults, which is due to the fact that the weak fault features of pitting–wear composite faults are not easily extracted by the traditional method, which in turn leads to poor identification results. For the traditional method, the feature extraction capability is limited because the loss function is constructed only based on the amplitude of the signal during the feature extraction process. It cannot extract features with small differences from signal to signal. However, we modified the loss function in the feature extraction process, and the essential information of the original data can be extracted not only by amplitude consistency but also by trend consistency, so the proposed CASAE method can solve the problem that the similarity between fault samples prevents the extraction of key subtle differences, which leads to low diagnostic accuracy. The proposed G-CASAE method adds the similarity loss function of labels in global fine-tuning after unsupervised pre-training to extract relevant features, which solves the problem of the misclassification of predicted labels due to the similarity of fault features and further improves the fault diagnosis accuracy of the model.

#### 4.1.3. Results and Analysis of Experiment 2

In order to verify the reliability of the algorithm under different amounts of data, Experiment 2 is designed by adding experimental samples. The superiority of G-CASAE is compared through learning and training with a larger sample size. In this stage, the experimental scenarios for Experiment 2 were set up for the gearbox data, as shown in Table 7.

The network models of Experiment 2 were the same as those of Experiment 1. The diagnostic results of each model are shown in Table 8.

It can be seen from columns 2 to 5 in Table 4 and Table 8 that only the number of training samples is increased. When the training sample size increases, it can effectively alleviate the overfitting problem of deep learning when the sample size is insufficient. However, it can be found that the robustness of the comparison model cannot be guaranteed, even if the sample is increased. In contrast, the proposed method still has high diagnostic accuracy on poor-quality test data. This further illustrates that the features extracted from different levels by the method proposed in this paper can better reflect the nature of the data.

Comparing Table 6 and Table 9, it can be obtained that the diagnostic accuracy can reach 86.12% when the proposed algorithm utilizes 500 samples at an SNR of 20dB. In order for the existing fault diagnosis methods to achieve consistent accuracy with the proposed method under the same noise, at least 2500 training samples would be required. In summary, when the existing fault diagnosis methods fail in the context of small samples with strong noise, the proposed method still has superior diagnostic accuracy that meets the practical engineering requirements.

To further present the diagnostic effect of the proposed method, the confusion matrix for each model in Experiment 2 is given in Figure 9.

As can be seen from Figure 9, the diagnostic accuracy of both the proposed method CASAE and G-CASAE is improved significantly when the samples are increased. This is because there are sufficient samples, more fault information is provided, and the effective feature extraction method can capture the fault features well, which in turn facilitates minor fault diagnosis.

### 4.2. Fault Diagnosis for the Rolling Bearing

#### 4.2.1. Dataset 2: Rolling Bearing Data

The bearing data collected from the test bench of the Case Western Reserve University rotary machinery vibration are used as the benchmark data to test the effectiveness of the proposed method [42]. The experimental platform is shown in Figure 10, which is mainly composed of a 1.5 kW (2 HP) motor, a torque transducer, and a dynamometer. In this experiment, test data were selected for the drive end bearing at a speed of 1772 rpm with a motor load of 1 HP, and a single point of failure of 0.007″ was simulated on the bearing by the electro-discharge machining technique. The sampling frequency was 12 kHz. Healthy monitoring data of the bearing were collected for H: normal, F_1_: inner ring fault, F_2_: outer ring fault, and F_3_: rolling ball fault, respectively. The collected vibration signals were used to construct the dataset with a sliding window of 400 and step size of 50.

In order to further verify the effectiveness of the G-CASAE algorithm, this experiment avoided the problem of insufficient model feature learning due to the small sample size during the construction of the bearing experimental samples. In this section, the sample set is constructed using a sliding window of size 400 for the preprocessing of the bearing data, and the experimental scenarios set up using the obtained samples are shown in Table 10.

#### 4.2.2. Results and Analysis of Experiment 3

To further verify the effectiveness and generalization performance of the proposed algorithm, Experiment 3 was used to verify the designed algorithm. The network model structures and parameters used in Experiment 3 are shown in Table 11. Simultaneously, we compared and analyzed our method with the above five methods and calculated the correct diagnosis accuracy for each type of fault. The experimental results are shown in the following Table 12.

From Table 12, the proposed G-CASAE method also has high diagnostic accuracy on the bearing dataset. Compared with the other five models, the diagnostic accuracy of the proposed method reaches 95.625%. Comparing columns 3 to 5 and column 7 in row 2, it can be seen that the incremental diagnostic accuracy of the improved feature extraction method in unsupervised pre-training can reach at least 10.4% compared with traditional deep learning methods such as LSTM. Comparing columns 6 and 7 in row 2, the proposed method is 4.75% more accurate than the feature fusion method. The comparison of columns 7 and 8 in row 2 shows that after effective unsupervised feature pre-training, by adding the cosine similarity measure in the supervised global fine-tuning process, the degree of tag similarity is taken into account in training, and the diagnostic accuracy of the proposed method can also be effectively improved.

In addition, from rows 2 to 5 of Table 12, it can be seen that all comparison methods are affected differently by noise of different intensities, and the proposed method has strong anti-interference ability and good generalization performance.

A comparative analysis of column 6 in row 6 and columns 7 to 8 in row 5 of Table 13 shows that when the bearing data are affected by the same noise, the fault diagnosis accuracy of the traditional method requires a large number of samples to achieve fault diagnosis accuracy comparable to the proposed method. Comparing Experiments 1, 2, and 3, it can be seen that the proposed algorithm is applicable to the identification and diagnosis of minor fault monitoring data of both gearboxes and rolling bearings. Moreover, the experimental results in different datasets maintain consistency, which shows that G-CASAE performs effective fault feature extraction and diagnosis in scenarios with small samples and strong noise.

The result further illustrates the superiority of the designed G-CASAE method. The confusion matrix of the diagnostic effect of each network model is shown in Figure 11.

It can be seen from Figure 11 that the proposed method is superior to the existing methods in the case that only 400 samples polluted with strong noise are available for training the DNN-based fault diagnosis model. This is because it can well discriminate F1 and F2, which are typical similar faults.

It can be also seen from Table 13 that 400 training samples polluted with strong noise are adequate for the proposed method to successfully train the DNN to achieve satisfactory fault diagnosis accuracy, while more than 2000 training samples are required for the traditional method to achieve comparative fault diagnosis accuracy.

## 5. Conclusions

Deep learning can be applied to the fault diagnosis of rotating machinery, where the diagnostic accuracy often depends on the number and quality of training samples. Since a small sample size of fault data polluted by strong noise is common in the field of engineering, it is necessary to develop a new training mechanism in order to accurately extract separable features suitable for discriminating similar faults in the event that only a small number of training samples polluted by strong noise is available. A trend feature consistency-guided deep learning method for minor fault diagnosis is proposed in this paper to make the DNN more powerful in feature presentation and similar fault discrimination. The proposed algorithm designs new loss functions for the accurate representation of data features guided by trend feature consistency and the accurate classification of faults guided by fault orientation consistency. The method overcomes the problem in which faults with similar membership cannot be effectively distinguished by traditional methods. Experimental validation of benchmark datasets shows that the proposed method is superior to traditional methods in the sense that more than a 10% diagnosis improvement can be achieved in the case in which only a small number of training samples polluted with strong noise is available. Compared with traditional methods, much fewer training samples are adequate for the proposed method to train a satisfactory DNN-based fault diagnosis model.

The experiment result also shows that when the training samples are polluted with strong noise, the fault diagnosis accuracy decreases, although it is still superior to the existing methods. In the case that the value of the training samples is too poor to establish a satisfactory fault diagnosis model, establishing a federation learning model to incorporate training data provided by different clients may be possible, which will be our future task.

## Figures and Tables

**Figure 1 entropy-25-00242-f001:**
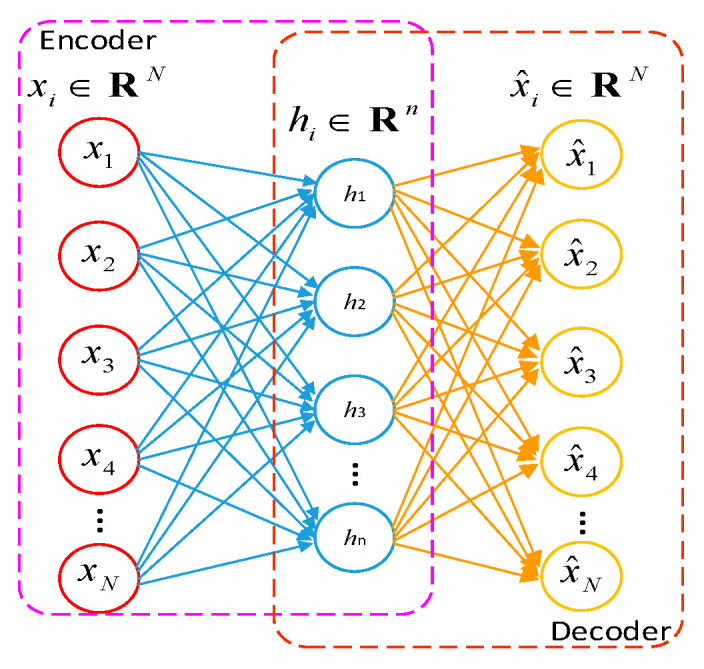
The structure of an auto-encoder.

**Figure 2 entropy-25-00242-f002:**
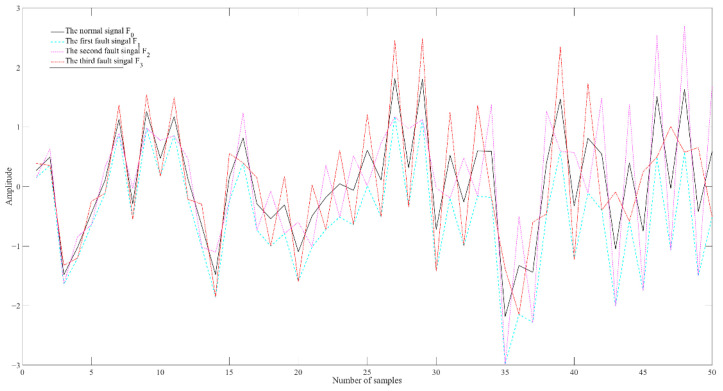
Distribution trend graph between input data and output data.

**Figure 3 entropy-25-00242-f003:**
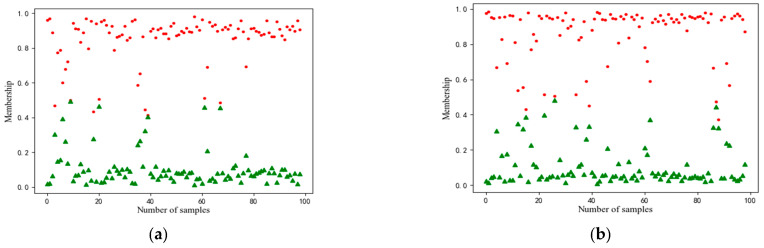
Visualization diagram of weak fault affiliation for different methods: (**a**) traditional deep learning minor fault diagnosis, (**b**) deep learning small fault diagnosis method based on denoising, (**c**) deep Learning for minor fault diagnosis-based correlation, (**d**) deep Learning minor fault diagnosis method-based fusion. Polka dot indicates the first degree of membership and triangle indicates the second degree of membership.

**Figure 4 entropy-25-00242-f004:**
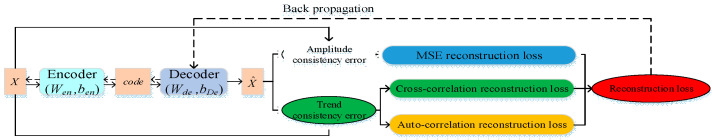
Schematic diagram of the proposed CASAE method.

**Figure 5 entropy-25-00242-f005:**
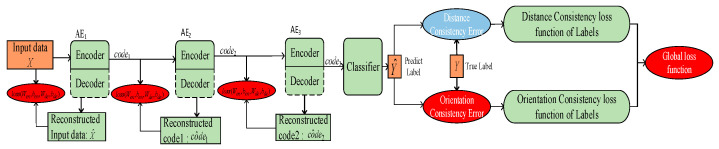
Schematic diagram of the proposed G-CASAE method.

**Figure 6 entropy-25-00242-f006:**
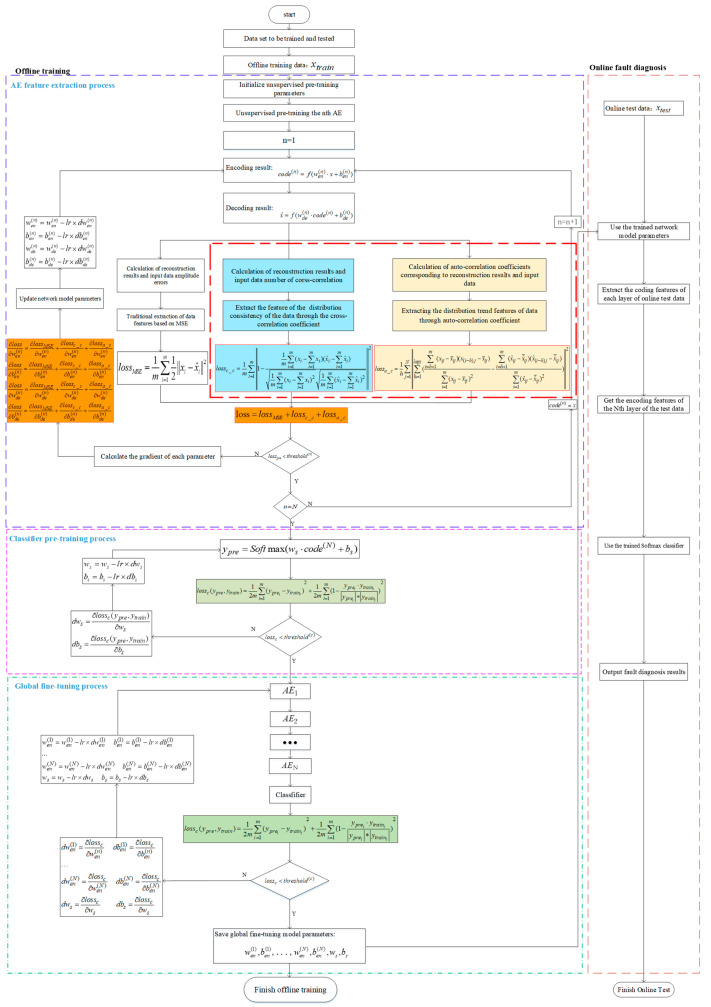
Flowchart of the proposed G-CASAE method.

**Figure 7 entropy-25-00242-f007:**
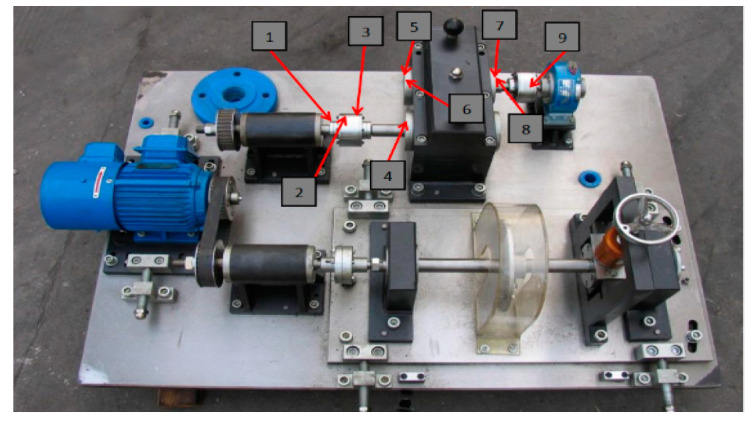
QPZZ-II rotary machinery vibration test bench.

**Figure 8 entropy-25-00242-f008:**
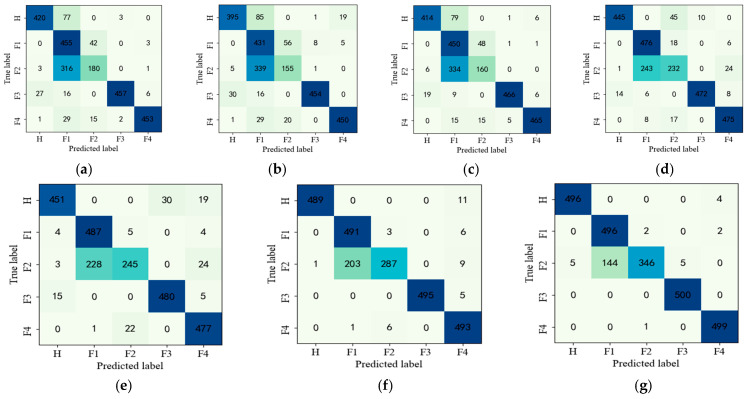
Confusion matrix for fault diagnosis results of parallel gearbox with no SNR for sample size of 100. (**a**) BP, (**b**) SAE, (**c**) SDAE, (**d**) LSTM, (**e**) MSFSAE, (**f**) CASAE, (**g**) G-CASAE.

**Figure 9 entropy-25-00242-f009:**
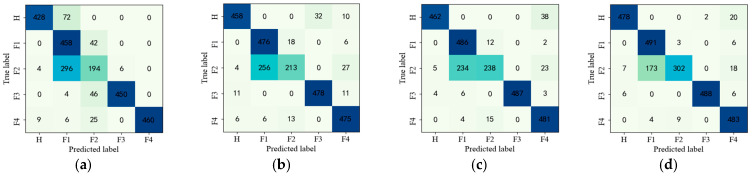
Confusion matrix for fault diagnosis results of parallel gearbox with no SNR for sample size of 500. (**a**) BP, (**b**) SAE, (**c**) SDAE, (**d**) LSTM, (**e**) MSFSAE (**f**) CASAE, (**g**) G-CASAE.

**Figure 10 entropy-25-00242-f010:**
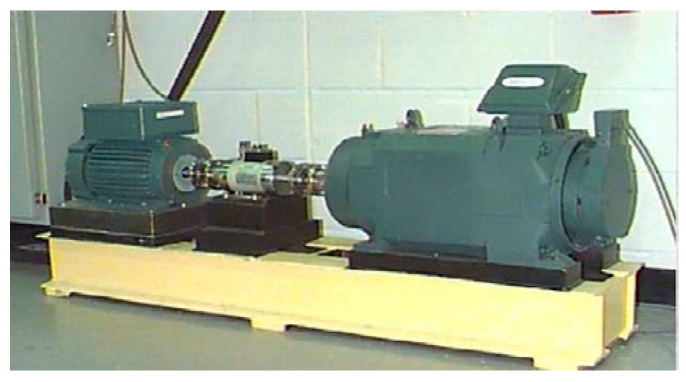
Bearing signal acquisition experimental platform.

**Figure 11 entropy-25-00242-f011:**
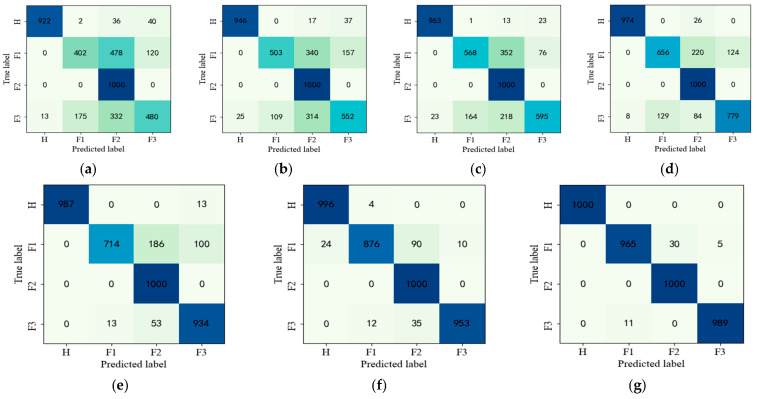
Confusion matrix for fault diagnosis results of rolling bearing with no SNR for sample size of 400. (**a**) BP, (**b**) SAE, (**c**) SDAE, (**d**) LSTM, (**e**) MSFSAE, (**f**) CASAE, (**g**) G-CASAE.

**Table 1 entropy-25-00242-t001:** Various sensor installation positions.

Channels	Serial Number in Figure 7	Sensor Location and Type
CH1	1	Photoelectric speed
CH2	2	Input shaft horizontal displacement
CH3	3	Input shaft vertical displacement
CH4	4	Input shaft left end cover vertical acceleration
CH5	5	Output shaft left end cover horizontal acceleration
CH6	6	Output shaft left end cover vertical acceleration
CH7	7	Output shaft right end cover horizontal acceleration
CH8	8	Output shaft right end cover vertical acceleration
CH9	9	Output shaft load side bearing magneto-electric speed

**Table 2 entropy-25-00242-t002:** Experiment 1 design for gearbox data.

Fault Type	Bearing Health Condition	Training Sample Size	Testing Sample Size
H	Normal	20	500
F_1_	Wear	20	500
F_2_	Pitting and wear	20	500
F_3_	Pitting	20	500
F_4_	Broken tooth	20	500

**Table 3 entropy-25-00242-t003:** The related fault diagnosis models for comparison.

Model	Model Description
BP	Backpropagation neural network
SAE	Stacked auto-encoder
SDAE	Stacked denoising auto-encoder
LSTM	Long short-term memory neural network
MSFSAE [26]	Multi-scale feature fusion stacked auto-encoder
CASAE	Cross-correlation and auto-correlation stacked auto-encoder
G-CASAE	Global fine-tuning of cross-correlation and auto-correlation stacked auto-encoder

**Table 4 entropy-25-00242-t004:** Network model parameters in gearbox diagnosis experiment.

Model	Model Parameters
BP	Number of layers: 5, neurons in each layer: 9/150/70/36/5, learning rate: 0.007
SAE	Number of layers: 5, neurons in each layer: 9/150/70/36/5, learning rate: 0.005
SDAE	Number of layers: 5, neurons in each layer: 9/150/70/36/5, learning rate: 0.005
LSTM	Cell number: 3, number of hidden neurons in the cell: 60, learning rate: 0.0004
MSFSAE	Number of layers: 5, neurons in each layer: 9/150/70/36/5, learning rate: 0.005
**CASAE**	Number of layers: 5, neurons in each layer: 9/150/70/36/5, learning rate: 0.005
**G-CASAE**	Number of layers: 5, neurons in each layer: 9/150/70/36/5, learning rate: 0.005

**Table 5 entropy-25-00242-t005:** Fault diagnosis results of different methods with different sample quality for 100 samples.

	Model	BP	SAE	SDAE	LSTM	MSFSAE	CASAE	G-CASAE
SNR								
None		78.70%	75.40%	78.20%	84.00%	85.60%	**90.20%**	**93.40%**
SNR:60		72.60%	70.40%	71.20%	79.20%	81.60%	**86.40%**	**90.48%**
SNR:40		58.48%	53.60%	56.40%	61.60%	67.20%	**84.80%**	**86.20%**
SNR:20		39.80%	38.20%	39.20%	49.60%	51.20%	**82.40%**	**83.70%**

**Table 6 entropy-25-00242-t006:** Diagnostic accuracy of the proposed method with 100 samples and traditional method with 1500 samples at SNR = 20 dB.

Model	BP	SAE	SDAE	LSTM	MSFSAE	CASAE	G-CASAE
Training sample size	1500					100	
Accuracy	64.40%	72.00%	74.6%	80.8%	83.4%	**82.40%**	**83.70%**

**Table 7 entropy-25-00242-t007:** Experiment 2 design for gearbox data.

Fault Type	Bearing Health Condition	Training Sample Size	Testing Sample Size
H	Normal	100	500
F_1_	Wear	100	500
F_2_	Pitting and wear	100	500
F_3_	Pitting	100	500
F_4_	Broken tooth	100	500

**Table 8 entropy-25-00242-t008:** Fault diagnosis results of different methods with different sample quality for 500 samples.

	Model	BP	SAE	SDAE	LSTM	MSFSAE	CASAE	G-CASAE
SNR								
None		79.60%	84.00%	86.16%	89.68%	92.40%	**96.76%**	**98.64%**
SNR:60		75.48%	80.28%	83.56%	80.00%	86.84%	**93.56%**	**94.16%**
SNR:40		60.72%	65.40%	72.00%	75.96%	77.72%	**89.72%**	**90.52%**
SNR:20		42.48%	46.20%	48.72%	54.12%	59.40%	**84.52%**	**86.12%**

**Table 9 entropy-25-00242-t009:** Diagnostic accuracy of the proposed method with 500 samples and traditional method with 2500 samples at SNR = 20 dB.

Model	BP	SAE	SDAE	LSTM	MSFSAE	CASAE	G-CASAE
Training sample size	2500					100	
Accuracy	66.80%	75.80%	76.60%	82.80%	86.40%	**84.52%**	**86.12%**

**Table 10 entropy-25-00242-t010:** Experiment 3 design for rolling bearing data.

Fault Type	Bearing Health Condition	Training Sample Size	Testing Sample Size
H	Normal	100	1000
F_1_	Inner ring	100	1000
F_2_	Outer ring	100	1000
F_3_	Rolling	100	1000

**Table 11 entropy-25-00242-t011:** Network model parameters in bearing diagnosis experiment.

Model	Model Parameters
BP	Number of layers: 5, neurons in each layer: 400/600/200/100/4, learning rate: 0.001
SAE	Number of layers: 5, neurons in each layer: 400/600/200/100/4, learning rate: 0.004
SDAE	Number of layers: 5, neurons in each layer: 400/600/200/100/4, learning rate: 0.004
LSTM	Cell number: 8, number of hidden neurons in the cell: 50, learning rate: 0.0004
MSFSAE [26]	Number of layers: 5, neurons in each layer: 400/600/200/100/4, learning rate: 0.004
**CASAE**	Number of layers: 5, neurons in each layer: 400/600/200/100/4, learning rate: 0.004
**G-CASAE**	Number of layers: 5, neurons in each layer: 400/600/200/100/4, learning rate: 0.004

**Table 12 entropy-25-00242-t012:** Fault diagnosis results of different methods with different sample quality for 400 samples.

	Model	BP	SAE	SDAE	LSTM	MSFSAE [26]	CASAE	G-CASAE
SNR								
None		70.10%	75.03%	78.15%	85.23%	90.88%	**95.63%**	**98.85%**
SNR:20		63.05%	72.88%	74.10%	82.93%	86.13%	**92.25%**	**93.13%**
SNR:10		55.70%	62.70%	70.65%	78.25%	79.75%	**87.65%**	**89.87%**
SNR:2		45.20%	58.53%	63.30%	69.85%	75.28%	**84.38%**	**86.70%**

**Table 13 entropy-25-00242-t013:** Diagnostic accuracy of the proposed method with 400 samples and traditional method with 2000 samples at SNR = 2 dB.

Model	BP	SAE	SDAE	LSTM	MSFSAE [26]	CASAE	G-CASAE
Training sample size	2000					400	
Accuracy	68.83%	74.62%	76.28%	83.53%	85.33%	84.38%	86.70%

## Data Availability

The data involved in this article have been presented in the article.

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
