# Peer review of "Trend Feature Consistency Guided Deep Learning Method for Minor Fault Diagnosis"

_entropy, 2023, doi:10.3390/e25020242_

Round 1
Reviewer 1 Report
This is an interesting work on the use of DNN in fault diagnosis. It is clearly in the thematic of the special issue "Fault Diagnosis Methods Based on Information Theory or Machine Learning: From Theory to Application". Some minor corrections could improve significantly the quality of the paper.
-In the abstract, could you precize DNN for readers not familiarized with this theory?
-Could you add a DNN reference at the beginning of section 2.1 ?
-The notations in section 2 must be uniformized. For example, in section 2.1, x_i is a set of vectors, at section 2.3, it is a vector, and then, at section 2.2, there is the notation x(i) that designates a vector. At section 2.3, precize the notation D(x_i) and x bar (it is the mean but you must say it). In (11), what are the notations y_pre, y_true and * ?
-Part 3.1 is interesting and convincing but could you explain what software, programs you used to make your simulations? If you write the programs, where can we find and test them?
-In section 3.1:
*precize in the title of Fig 4, what does CASAE mean?
* Wen, ben after equatio (13) should be precized later, before (15) and (18).
*you have 2 equations with number (15) and the reference (22) must be replaced by the reference (13) after Iossa_c (equation 15).
-Section 3.3.2, replace yraal/yrael by yreal.
-I do not understand the beginning of section 3.3.3. Could you explain better Feature_CF?
-In figure 8, you must indicate in the title what does the axes, the numbers, the colours... represent.
Reviewer 2 Report
It is opinion of the reviewer that the paper presents interesting technique, where an original and innovative work is shown with a right methodology, properly organized and structured, and the authors have worked exhaustively, taking care of the technical details and obtaining good results, classifying data from two vibration databases.
However, it is opinion of the reviewer that, among others, some suggestions could improve the paper:
- The introduction section should be improved and more references should be added. This section should take into consideration the most relevant studies on the subject to build a complete scientific framework. The paper does not include all relevant references to the topic: it is good in Chinese authors but in the rest it is deficient.
- The text should be revised and some errors corrected.
- It would be advisable to complete the list of acronyms.
- Furthermore, It would be advisable that acronyms do not appear in the Abstract.
For these reasons, the reviewer proposes to publish the manuscript after minor revision.
